# Primary Biliary Cholangitis: Immunopathogenesis and the Role of Bile Acid Metabolism in Disease Progression

**DOI:** 10.3390/ijms26167905

**Published:** 2025-08-16

**Authors:** María Del Barrio, Álvaro Díaz-González, Marta Alonso-Peña

**Affiliations:** 1Gastroenterology and Hepatology Department, Clinical and Traslational Research in Digestive Diseases, Valdecilla Research Institute (IDIVAL), Marqués de Valdecilla University Hospital, 39008 Santander, Spain; maria.delbarrio@scsalud.es (M.D.B.); alvaro.diaz@scsalud.es (Á.D.-G.); 2Departamento de Anatomía y Biología Celular, Universidad de Cantabria, 39011 Santander, Spain

**Keywords:** primary biliary cholangitis, bile acids, cholestasis, autoimmune disease

## Abstract

Primary biliary cholangitis (PBC) is a chronic, immune-mediated liver disease characterized by progressive destruction of the small intrahepatic bile ducts, leading to cholestasis, inflammation, and ultimately fibrosis and cirrhosis. This review emphasizes the central role of bile acids in PBC pathogenesis, exploring how disruptions in their synthesis, transport, and detoxification contribute to cholangiocyte damage and disease progression. In addition to discussing the autoimmune features of PBC, including the presence of specific autoantibodies and cellular immune responses, we examine how bile acid dysregulation exacerbates cholestasis and promotes lipid metabolic disturbances. Particular attention is given to the “bicarbonate umbrella” hypothesis, which describes a protective mechanism by which cholangiocytes resist bile acid–induced injury—an essential factor disrupted in PBC. The aim of this review is to summarize current knowledge gaps in the pathophysiology of PBC, with a focus on the role of bile acids not only as key drivers of disease mechanisms, but also as potential biomarkers of disease progression and treatment response.

## 1. Introduction

Primary biliary cholangitis (PBC) is a rare, chronic cholestatic liver disease that predominantly affects middle-aged women [1]. If not treated properly, it can lead to liver cirrhosis and hepatocellular carcinoma. It is diagnosed when two of the following three criteria are met: chronic elevation of alkaline phosphatase (ALP), positivity for anti-mitochondrial antibodies (AMA) or other specific autoantibodies and/or compatible liver biopsy [2,3]. Despite being considered a rare disease, its prevalence and incidence are increasing [4].

Most patients with PBC are asymptomatic at the time of diagnosis [3]. Currently, the most reported symptoms are fatigue, pruritus, and discomfort in the right upper quadrant. Although classical signs and symptoms of the disease—such as jaundice, xanthomas, or xanthelasmas—have been historically described in association with cholestasis, their frequency appears to be declining, likely due to earlier diagnosis and/or timely initiation of treatment. Moreover, they may associate other autoimmune diseases, being Sjögren syndrome the most prevalent one [2,3], although it is sometimes misdiagnosed due to sicca symptoms (dryness of the mouth, eyes, and other mucous membranes) commonly seen in PBC [5]. CREST (calcinosis, Raynaud’s phenomenon, esophageal dysfunction, sclerodactyly and telangiectasia) syndrome is also common in patients with PBC [6,7].

First line treatment is ursodeoxycholic acid (UDCA), a hydrophilic bile acid (BA), that had demonstrated to increase transplant-free survival. Regrettably, approximately 40% of PBC patients exhibit an incomplete response [8]. Thus, second line treatments are necessary. Three drugs have been approved so far: Obeticholic acid (OCA), which was later revoked by the European Commission [9], Elafibranor [10] and Seladelpar [11].

Despite the remarkable therapeutic advances witnessed in recent years in the field of PBC, many uncertainties and unresolved aspects remain. In this review, we aim to elucidate the roles of the immune system and BAs in the progression of the disease, with the goal of gaining a deeper understanding of underlying mechanisms and identifying the key questions that still need to be addressed. To this end, we based our conclusions on the results reflected in the scientific literature, obtained by searching the bibliographic databases Pubmed, Web of Science and Scopus.

## 2. Pathophysiology of PBC

The exact etiopathogenesis of PBC remains unknown; however, it is believed to involve a complex interplay between immune-mediated mechanisms and biliary epithelial pathways. This chronic immune–biliary interaction leads to progressive destruction of the intrahepatic bile ducts, resulting in cholestasis, inflammation, and ultimately liver fibrosis [3].

### 2.1. Autoimmunity

#### 2.1.1. Autoantibodies in PBC

The most common autoimmune marker in PBC is AMA, found in 90–95% of PBC patients. This antibody has reactivity to the inner mitochondrial membrane and the mitochondrial fraction containing succinate dehydrogenase. The M2 antigen, a subunit of this complex, is considered highly specific for PBC. When assessed by indirect immunofluorescence (IIF), AMA demonstrates both high sensitivity (≈85%) and specificity (≈99%) [12]. However, the role of AMA as an etiologic agent or as an epiphenomenon is still under debate. It is known that not all patients with positive AMA present clinical features of PBC: frequency of AMA-M2 positivity in general population is set around 0.1–0.2% [3,13,14]. In fact, only one in six patients with positive AMA will develop clinical PBC within the next 5 years [15,16]. Moreover, AMA titre does not correlate with disease severity neither symptom, and it can fluctuate during follow up, unrelated to response to the treatment [17]. It should also be mentioned that AMA positivity has been observed in patients with extrahepatic diseases, for example, in lupus erythematosus, systemic sclerosis, Sjögren syndrome, inflammatory myositis or chronic graft versus host disease after allogenic stem cell transplantation [17,18].

Antinuclear antibodies (ANA) are detected in approximately 50% of patients. The most common IIF pattern of ANA are the nuclear-rim dots pattern, which react to the nuclear pore complex protein gp210, and multiple nuclear dots, which react to the nuclear protein sp100. Both of them are very specific (97–98%) but less sensitive (<30%) [12]. What is more, anti-gp210 and anti-sp100 are also prognostic markers, as PBC patients positive for those antibodies had worse responses to UDCA treatment and worse prognosis [19,20].

Additionally, autoantibodies against kelch-like protein 12 (KLHL12) and hexokinase 1 (HK-1) have emerged as useful diagnostic tools, particularly in AMA-negative PBC patients, in whom they are detected in 10–35% of cases. Both markers demonstrate high specificity (>90%) and moderate sensitivity (~70%) [12,20,21].

#### 2.1.2. Cellular Immune Responses

The histopathology of PBC is characterized by portal and periportal inflammatory infiltrate surrounding intrahepatic bile ducts. This infiltrate predominantly comprises CD4+ and CD8+ T lymphocytes, macrophages, B lymphocytes, plasma cells, natural killer (NK) cells, natural killer T (NKT) cells, and a variable presence of eosinophils [22]. Although the precise trigger for the breakdown of immune tolerance remains unclear, activated cholangiocytes appear to play an active role in biliary injury by contributing to the recruitment and modulation of the immune response [22].

T lymphocytes are pivotal in the autoimmune response in PBC [23]. Current evidence supports that the activation of CD8+ T cells within the inflamed liver microenvironment, prompted by autoreactive CD4+ T cell activation, results in significant proinflammatory cytokine production and the destruction of cholangiocytes via direct cytotoxic effects [24]. It is noteworthy that early-phase PBC is characterized by an interleukin-12 -driven Th1 immune response, whereas an interleukin-23 -mediated Th17 response predominates in later stages [22].

Moreover, regulatory T cells (Treg) are found to be reduced in the liver and peripheral blood of patients with PBC, likely due to the immunosuppressive effect of B cells, which are also responsible for the aberrant production of autoantibodies [24].

Differences in cellular immune response have been also observed between UDCA responders and non-responders. Transcriptomic profiling of monocytes, Th1, Th17, Treg, and B cells from the peripheral blood of patients with PBC under treatment has shown that UDCA responders have a stronger regulation of immune responses, while monocytes exhibit a proinflammatory phenotype and Th1, Th17 and Treg are activated in non-responders [25].

### 2.2. Cholestasis

Cholestasis refers to the impairment or cessation of bile flow through the biliary system, resulting in the accumulation of bile constituents, particularly BAs, within the liver. BAs are major components of bile, along with cholesterol and phospholipids, and play essential roles in lipid digestion and signalling [26]. Cholestasis can be classified as intrahepatic—where BAs accumulate within hepatocytes, typically due to genetic disorders affecting bile formation or export—or associated with cholangiopathies, in which the obstruction or dysfunction of intrahepatic or extrahepatic bile ducts leads to impaired bile flow [27]. In cholestatic liver diseases such as PBC, the accumulation of BAs induces hepatocellular and cholangiocellular injury. Consequently, disrupted BA metabolism, altered lipid handling, and progressive hepatobiliary damage are central to the pathogenesis and progression of PBC (Figure 1).

#### 2.2.1. Altered Bile Acid Metabolism

Since the early 1960s, the hepatotoxic effects of BAs have been the subject of investigation. Hydrophobic BAs are more toxic due to their detergent properties on cellular membranes, thereby damaging both cholangiocytes and hepatocytes. On the contrary, hydrophilic BAs are innocuous [28]. Primary BAs are less toxic than secondary BAs, whereas conjugated BAs are also less toxic than unconjugated ones [29,30] (Figure 2). Nonetheless, early studies found that BAs at physiological concentrations induce hepatic stellate cell and fibroblast proliferation and activation in vitro and animal models [31]. Therefore, any bile leakage from the bile duct could initiate a fibrotic response in the liver. Thus, keeping BA metabolism within its physiological state is essential for the correct function of the hepatobiliary system, being Farnesoid X Receptor (FXR) the orchestrator of BA homeostasis (For a detailed review on the topic, see Fuchs et al. [26]).

A growing body of evidence reveals that significant alterations in the composition, conjugation, and systemic distribution of BAs are hallmarks of early and untreated PBC, with important implications for disease pathogenesis and therapeutic monitoring. In untreated PBC patients, total serum BA concentrations are significantly increased compared to non-cholestatic controls, with a predominant accumulation of primary BAs. Trottier et al. reported a 13.5-fold increase in primary BAs in PBC serum, primarily driven by taurocholic acid (TCA) accumulation, which alone constituted 52% of total serum BAs versus 22% in controls (*p* = 0.0047) [32]. Furthermore, taurine-conjugated BAs, including TCA, taurochenodeoxycholic acid (TCDCA) and taurodeoxycholic acid (TDCA), represented approximately 50% of total BAs in PBC, compared to only 14% in controls. Conversely, the proportion of glycine conjugates remained relatively stable, although glycocholic acid (GCA) increased 19-fold, and glycochenodeoxycholic acid (GCDCA) was identified as the second most abundant BA in serum [32]. Tang et al. confirmed these findings, identifying GCDCA, GCA, glycodeoxycholic acid (GDCA), glycolithocholic acid (GLCA), TCDCA, and TCA as significantly elevated in untreated PBC serum [33]. Chen et al. also demonstrated elevated serum concentrations of GCA, TCA, and GCDCA in naïve PBC patients. Notably, the ratio of conjugated to unconjugated BAs was markedly increased in both serum and feces of naïve PBC patients, indicating a systemic shift toward BA conjugation [34].

**Figure 2 ijms-26-07905-f002:**
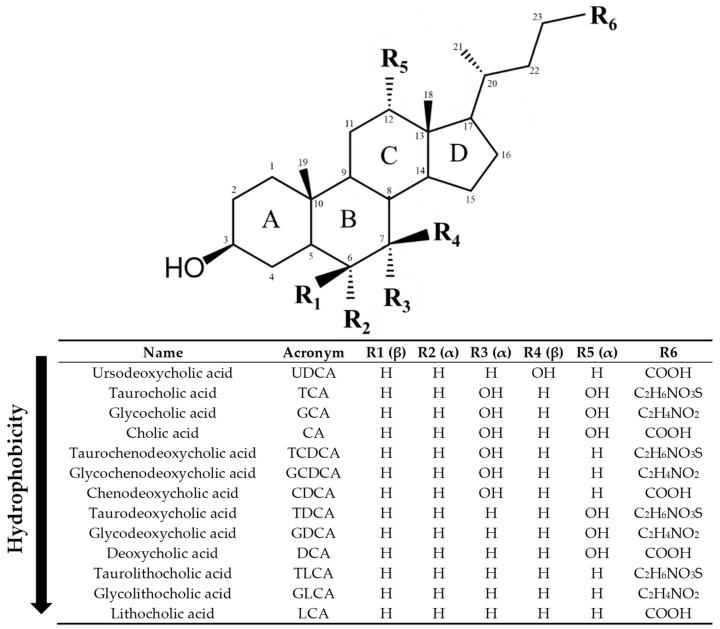
Structure of bile acids (BAs). Ursodeoxycholic acid (UDCA) is a tertiary BA in humans, best known due to its pharmacological actions. Cholic acid (CA) and chenodeoxycholic acid (CDCA) are primary BAs in humans, whereas deoxycholic acid (DCA) and lithocholic acid (LCA) are secondary BAs. Primary and secondary BAs can undergo conjugation with taurine and glycine, leading to the corresponding conjugated forms (TCA, GCA, TCDCA, GCDCA, TDCA, GDCA, TLCA, GLCA). Hydroxyl groups (OH) in α-orientation are located below the steroid core and are axial with respect to the plane of the steroid core. Hydroxyl groups in β orientation are located above the steroid core and are equatorial to it. The BAs have been ordered from lowest to highest hydrophobicity according to octanol/water partition coefficient predictions from ChemDraw 19 software and literature [29,30,35].

The dysregulation of BA metabolism in PBC extends to the balance between primary and secondary BAs (Figure 1). Under normal physiological conditions, primary BAs, synthesized from cholesterol in the liver, are secreted into the intestinal lumen where they undergo deconjugation and 7α-dehydroxylation by anaerobic gut bacteria, yielding secondary BAs. These reactions are predominantly catalyzed by bacterial species within the *Clostridium* genus, notably *Clostridium scindens*, which harbor the bile acid-inducible (bai) operon necessary for 7α-dehydroxylation [36,37]. This microbial transformation is essential for BA diversification and maintaining a balanced BA pool. Reduced levels of deoxycholic acid (DCA), GDCA, lithocholic acid (LCA), and taurolithocholic acid (TLCA) in serum from PBC patients reflect decreased microbial biotransformation in the gut, likely due to altered enterohepatic circulation and cholestatic impairment [34]. This biochemical disruption parallels several alterations in the gut microbiota of PBC patients [38]. Tang et al. reported reduced diversity and compositional changes in the intestinal microbiome, including a decreased abundance of *Ruminococcaceae*—a family involved in secondary BA generation [33]. These microbial shifts correlate with the observed biochemical patterns and suggest that microbial dysbiosis contributes to the secondary BA deficiency characteristic of PBC. This shift results in a more hydrophilic and less cytotoxic serum BA composition, albeit reflective of a pathological adaptation rather than effective homeostasis. However, secondary BA deficiency is not a benign feature. Secondary BAs, particularly DCA and LCA, activate nuclear receptors such as FXR and G protein-coupled BA receptor (TGR5), which modulate inflammatory responses and intestinal barrier integrity [39]. Sinha et al. have shown that reduced secondary BA levels promote intestinal inflammation by disrupting FXR-mediated immune regulation, further implicating BA–microbiota interactions in the progression of chronic liver diseases [40]. Thus, the dysbiosis-induced deficiency of secondary BAs in PBC may exacerbate hepatic injury via immune dysregulation and increased permeability of the intestinal barrier [38,41].

#### 2.2.2. Dysregulation of Lipid Metabolism

Dyslipidaemia is present in up to 70% of patients with PBC [3]. As mentioned above, BA transport and metabolism can be altered, which has implications for the solubilization and excretion of cholesterol, due to their role in dietary lipid absorption but also because of their hormone-like signalling properties [42]. Thus, BA are key molecules in the regulation of lipid metabolism. Through the activation of FXR, BAs stimulate the expression of Peroxisome Proliferator-Activated Receptor (PPAR) α, thereby promoting fatty acid (FA) β-oxidation and contributing to a reduction in serum triglyceride (TG) levels in humans [43,44]. Moreover, FXR activation promotes *SHP* expression, which inhibits apolipoprotein B (APOB) and thereby TG export from hepatocytes, resulting in reduced formation of very low-density lipoprotein (VLDL) particles [45]. Fibroblast growth factor 19 (FGF19) induces a reduction in mitochondrial acetyl-coenzyme A-carboxylase 2, promoting FA oxidation and inhibiting FA synthesis, lowering liver fat accumulation and plasma TG levels [46]. FXR also facilitates VLDL catabolism and enhances FA uptake by skeletal muscle and adipose tissue [26]. Plasma high-density lipoprotein (HDL) cholesterol uptake is also increased by FXR [47] (Figure 1). Consequently, in the plasma of PBC patients there may be alterations in phospholipids, TG, and HDL, LDL, and VLDL lipoproteins [48]. An increase in X lipoprotein (Lp-X) has been described in patients with cholestasis. Lp-X lacks APOB in its structure, which makes the union to liver LDL receptors impossible, stimulating the elimination of excess BAs and cholesterol from the systemic circulation. In addition, it has a density similar to that of LDL, so in patients with PBC, elevated LDL cholesterol levels must be interpreted carefully to avoid misdiagnosis [46].

Despite mentioned dyslipidaemia, the increase in cardiovascular risk (CVR) in patients with PBC is still a controversial topic with variable results in the literature. In a Mexican retrospective study that included 54 patients with PBC, it was shown that they did not have a higher risk for cardiovascular events (CVE) at 10 years, comparing with healthy controls (5.3% vs. 4.1% *p* = 0.723), even if they had higher levels of total cholesterol, LDL and TG [49]. A large Italian study, with 400 PBC patients, also found that PBC patients did not have more CVE and that higher cholesterol levels were not a risk factor for CVE [50]. Solaymani-Dodaran et al. concluded that PBC patients did not have a higher risk for heart attack, stroke, neither transient ischemic attack compared with a general population control group [51]. On the other hand, Allocca M et al. demonstrated that older age PBC patients and those with arterial hypertension (AHT) had an increased risk of subclinical atheromatosis, regardless of having hypercholesterolemia [52]. Along these lines, another Italian study observed that 32% of PBC patients had a metabolic syndrome, which contributed to this subgroup suffering a greater number of CVEs [53]. Therefore, the evaluation of cofactors that increase CVR seems important. Finally, a novel study performing GWAS (genome-wide association study) found an association between genetically predicted PBC and the risk of developing diabetes, acute myocardial infarction, AHT, atrial fibrillation, heart failure, and stroke. However, it did not find a causal relationship with atherosclerosis or coronary artery disease [54].

Nevertheless, the literature presented is heterogeneous, with most studies having a small sample, so future studies are needed to clarify the CVR in these patients.

#### 2.2.3. Biliary Damage

PBC is characterized by chronic, non-suppurative, lymphocytic cholangitis targeting the small intrahepatic bile ducts [55]. Histological damage in the portal zone affecting cholangiocytes is common [56]. Therefore, liver histology is useful for the staging and prognosis of PBC. In this regard, several classification systems have been proposed, with Ludwig and Scheuer’s classification as well as Nakanuma classification being the most used in clinical settings [3]. Both systems consider portal inflammation, bile duct lesions and fibrosis. Dense periductal lymphocytic infiltrates associated with the destruction of bile duct epithelium, known as florid duct lesions, are characteristic of early-stage PBC and include T lymphocytes, macrophages, and occasional eosinophils and epithelioid granulomas [3]. This leads to ductopenia, ductular reaction (i.e., proliferation and expansion of dysfunctional bile duct-like structures) and portal expansion which can be highlighted by keratin 7 immunostaining [3,56]. Ductular reaction correlates with fibrosis stage in PBC and inversely correlates with UDCA response [31]. If disease progresses, inflammatory activity extends beyond the limiting plate of the portal area and it is sometimes associated with ballooning degeneration and emperipolesis [3]. In later stages, fibrosis progresses to bridging fibrosis and eventually to cirrhosis, characterized by broad fibrous septa and regenerative nodules [3]. However, nodular regenerative hyperplasia of the liver can be found in early histological stages of PBC [57]. Finally, orcein-positive granules are indicative of copper-binding protein accumulation due to chronic cholestasis [3].

Biliary injury contributes to the progression of PBC through multiple interconnected mechanisms. As the disease advances, apoptotic processes increasingly surpass compensatory cholangiocyte proliferation, promoting progressive bile duct loss. Moreover, apoptosis itself facilitates fibrogenesis, as apoptotic bodies are believed to stimulate hepatic stellate cell activation [58]. Importantly, during cholangiocyte apoptosis, the principal mitochondrial autoantigen PDC-E2 remains structurally intact within apoptotic blebs (Figure 3). These blebs provoke a pronounced pro-inflammatory cytokine response in granulocyte-macrophage colony-stimulating factor–primed mononuclear cells from PBC patients in the presence of AMAs, further amplifying local immune-mediated tissue damage [59].

Recent evidence also indicates that cholangiocyte senescence—a state of cell cycle arrest characterized by apoptosis resistance and abundant secretome—contributes to the pathogenesis of PBC [60] (Figure 3). Expression of senescence markers p21 and p16 is increased in small bile ducts in PBC patient tissue, being more extensive in later stages of PBC compared to early stages. Besides, p16 expression in bile ducts significantly correlated to failed response to UDCA treatment [61]. These cells exhibit a pronounced senescence associated secretory phenotype, characterized by elevated expression of chemoattractant, pro-inflammatory and profibrotic mediators such as CCL20, interferon and TGF-β, promoting immune cell infiltration, sustained inflammation and fibrogenesis [31,60].

Moreover, cholangiocytes express cell-surface adhesion molecules that allow adhesion and recognition of lymphocytes and they act as antigen presenting cells by expressing HLA class II and accessory molecules that are responsible for the costimulatory signal to T cells [58].

### 2.3. Integrative Theories: The Bicarbonate Umbrella Hypothesis

Cholangiocytes are polarized cells, which secrete bicarbonate (HCO_3_) and water. Furthermore, they generate up to 30% of the total bile flow in humans although representing a small proportion of total liver cells [62]. This HCO_3_ secretion takes place via the Cl^−^/HCO_3_ exchanger AE2 located at the apical membrane of cholangiocytes. Besides, membrane-bound carboanhydrase may spread the HCO_3_ [63]. The expression of AE2 is regulated by hormonal factors: in the postprandial period, secretin (Sct) promotes an increase in the intracellular flow of cAMP, which stimulates the transport of vesicles containing AE2, the Cl channel cystic fibrosis transmembrane regulator (CFTR), and the water channel aquaporin 1, to the luminal membrane. It is important to note that CFTR creates a chloride gradient that promotes Cl^−^/HCO_3_ exchange [22]. It has recently been proven that biliary Sct/Sct receptor/CFTR/AE2 expression and bile HCO_3_ levels were reduced in late-stage PBC mouse models and human samples. Furthermore, Sct treatment restores bicarbonate umbrella and decreased bile duct loss, ductular reaction and liver inflammation in mouse models [64].

Acidification of bile at the apical membrane promotes the persistence of glycine-conjugated BAs which can penetrate into cholangiocytes, resulting in oxidative stress, apoptosis and cytotoxicity [22]. Furthermore, AE2 deficiency alters cholangiolar intracellular pH stimulating the activation of the HCO_3_ sensor soluble adenylyl cyclase what increases the apoptosis induced by BAs [65]. Thus, it is hypothesized that HCO_3_ secretion protects cholangiocytes’ apical surface by maintaining an alkaline pH barrier or “umbrella” on the outer surface of the apical membrane, keeping bile salts in a polar state [22]. Moreover, UDCA promotes the alkaline barrier by increasing HCO_3_ secretion. It is also known that most patients respond to UDCA therapy, improving cholestasis and having a better prognosis than non-responders [66]. This fact supports the idea that AE2 deficiency and a defective HCO_3_ umbrella are key determinants of bile duct damage in PBC [22].

As mentioned before, once cholangiocytes are damage, they become active participants in disease progression. Importantly, PDC-E2 is overexpressed by stressed cholangiocytes, thus enhancing the autoimmune response and playing a crucial role in recruiting immune cells [22]. In fact, CD8+ T cells depend on AE2 to maintain the intracellular pH homeostasis [67]. Additionally, decreased expression of AE2 is correlated with dysregulated autophagy, expression of PDC-E2, and senescence [68].

However, the injury to cholangiocytes exemplifies a complex triangular relationship among immunity, cholangiocytes, and cholestasis. In fact, the current debate focuses on determining whether the primary trigger is a defect in the bicarbonate “umbrella” or a disruption of biliary homeostasis as a result of immune dysregulation [69] (Figure 3).

## 3. Bile Acids in the Treatment of PBC

Importantly, the pharmacology action of all current therapeutic options for managing PBC involves the manipulation of BA metabolism (Figure 1), although some drugs also target other mechanisms. UDCA, the first-line therapy for PBC, induces profound changes in circulating BA profiles. Administered orally, UDCA exerts its principal pharmacological effects in PBC through its choleretic properties (enhancing bile flow) and its antioxidant activity [70]. Following UDCA administration, the BA pool becomes enriched with hydrophilic species—mainly UDCA and its tauro- and glycoconjugates (TUDCA and GUDCA)—at the expense of endogenous primary and secondary BAs [34,71,72]. Dilger et al. demonstrated that therapeutic efficacy is associated with achieving a plasma concentration of UDCA derivatives of at least 75%, which corresponds to a biliary enrichment of ≥40% [72]. Treatment with UDCA also normalizes the excessive taurine conjugation observed in untreated PBC. Taurine conjugation of chenodeoxycholic acid (CDCA), which is disproportionately elevated in naïve PBC, was significantly reduced following UDCA therapy, restoring a conjugation pattern more closely resembling that of healthy controls [72]. Moreover, Tang et al. showed partial restoration of gut microbial composition following UDCA treatment, suggesting that pharmacological modulation of BAs may exert secondary effects on the microbiome [73].

Beyond UDCA, OCA was the first second-line treatment approved for PBC in patients who do not respond to UDCA. However, its authorization was revoked by the European Medicines Agency [9], based on the results of the COBALT study [74]. The COBALT trial evaluated the impact of OCA on clinical outcomes in patients with PBC. The authors reported that the primary composite endpoint occurred at similar rates in both the OCA and placebo groups [74]. However, it is highly relevant to note that the interpretation of these findings was substantially confounded by functional unblinding and crossover to commercially available OCA therapy, particularly among patients in the placebo arm. This significantly compromised both the quality of the data and the interpretability of the trial’s results. When analyses were adjusted using inverse probability of censoring weighting and as-treated methods, the hazard ratios shifted in favor of OCA [74]. OCA is a semi-synthetic BA analogue and potent FXR agonist, exerting its therapeutic effects by modulating key steps in BA homeostasis. FXR activation downregulates hepatic BA synthesis via repression of cholesterol 7α-hydroxylase (CYP7A1) and enhances BA excretion by upregulating bile salt export pump (BSEP) and inducing FGF19, which suppresses CYP7A1 expression in a gut-liver endocrine loop [62]. A randomized, placebo-controlled PET study using [11C]-cholylsarcosine in UDCA-treated PBC patients demonstrated that OCA significantly increased canalicular secretion of conjugated BA from hepatocytes, as indicated by a 73% increase in the secretion rate constant and a 30% reduction in intracellular residence time of the tracer [75]. These findings suggest a direct cytoprotective mechanism via acceleration of BA efflux from hepatocytes, thereby reducing exposure to cytotoxic BA.

Fibrates, used for long as off-label treatments for PBC, are PPAR agonists with pleiotropic effects on lipid metabolism, inflammation, and BA regulation. In the BEZURSO trial, Bezafibrate, a pan-PPAR agonist, induced biochemical normalization in 31% of PBC patients who did not respond to UDCA [76]. Moreover, the FITCH trial showed that Bezafibrate is useful in treating severe itch in patients with fibrosing cholangiopathies [77]. Pharmacological activation of PPARα induces phospholipid transporter expression, facilitating the formation of mixed micelles, downregulates BA synthesis and upregulates BA glucuronidation and canalicular transport [78]. Besides, PPARδ activation in mice has shown to suppress BA synthesis through the induction of Fibroblast growth factor 21 (FGF21) and subsequent downregulation of CYP7A1 [79] and promote cholesterol excretion into bile [80]. In the context of PBC, fibrates have specifically demonstrated to cause enhanced biliary phospholipid secretion and inhibition of BA synthesis [3,81].

As mentioned, Seladelpar, a specific agonist of PPARδ, and Elafibranor, a dual agonist of PPARα/δ, have been approved as second line treatments for PBC [10,11]. Elafibranor has been shown to be effective in a phase III trial conducted in PBC patients classified as non-responders to UDCA, with 51% of treated patients meeting the primary end point (defined as an ALP level < 1.67 times the upper limit of normal [ULN], with a reduction of ≥15% from baseline, and total bilirubin at or below the ULN) versus 4% in the placebo group, and 15% showing complete normalization of ALP [78,82]. In phase III trial, Seladelpar also met the primary end point (the same as in the Elafibranor trial) in 61.7% patients versus 20% in the placebo group, and up to 25% normalizing ALP. Plus, Seladelpar also significantly improved pruritus [83,84,85]. These results have led to FDA accelerated approval for both drugs [10,11]. Furthermore, both drugs have demonstrated a significant impact on BA metabolism. For instance, Seladelpar significantly reduced serum BAs [86,87], whereas Elafibranor significantly reduced circulating levels of the BA precursor 7α-hydroxy-4-cholesten-3-one (C4), showed a trend to reduce secondary BA levels with no apparent effect on primary BA levels [78,82].

Finally, pharmacological inhibition of apical sodium-dependent BA transporter (ASBT) has been shown to be effective in cholestasis, especially in the management of pruritus [88,89,90]. Linerixibat, a targeted inhibitor of ASBT has recently proved effectiveness in the management of pruritus in PBC patients in a phase III trial [88]. Beyond itch, already addressed above, Hegade et al. observed a significant reduction in all glyco- and tauro-conjugated BAs after two weeks of therapy in a cohort of PBC patients with pruritus treated with Linerixibat, although the decrease in total BA levels did not reach statistical significance. Interestingly, CDCA and DCA levels increased post-treatment, while cholic acid (CA) remained unchanged, suggesting differential modulation of primary BA synthesis pathways [91]. Moreover, results from a phase IIb trial in PBC patients have shown that Linerixibat treatment significantly decreases total serum BAs and increases C4 but reaching a plateau after 4 weeks [92].

## 4. Bile Acids as Biomarkers of PBC Progression and Therapy Response

Several studies have explored the role of BA as biomarkers for the diagnosis and prediction of progression and therapy response in PBC. Liu et al. [93] analyzed 15 species of BAs by liquid chromatography coupled with mass spectrometry and evaluated their diagnosis capacity compared to healthy subjects by the area under the receiver operating characteristic curves (AUROC), specificity and sensitivity. They found up to 8 BAs species with AUROC values between 0.827 and 0.992, specificity ranging from 80 to 100% and sensitivity 80 to 90%, and the best AUROC was for DCA, with a sensitivity and specificity of 90% [93]. However, it is worth mentioning that these excellent results might be biased by the small number of subjects included in the study.

Besides, BAs have been proposed as differential diagnosis biomarkers. In this regard, LCA and TLCA could help in differentiating PBC from Autoimmune Hepatitis [94]. In fact, Sang et al. performed a comprehensive study evaluating the BA profile in chronic liver diseases of different etiologies [95]. They included 719 patients with chronic liver disease (hepatitis B virus [HBV], hepatitis C virus [HCV], nonalcoholic steatohepatitis [NASH], alcohol-induced liver disease, and PBC) and 153 healthy controls and quantified 21 serum BAs using ultra-performance liquid chromatography and triple quadrupole mass spectrometry. Importantly, the authors emphasized that, considering potential detection biases, inter-individual variability, and prior evidence, they focused the analysis on proportion and ratio variables rather than absolute concentrations. They found that total BA levels were significantly higher in PBC than HBV, HCV, and NASH groups and they showed the lowest ratio of primary to secondary BAs. Moreover, the percentage of TCA, TCDCA and the ratio conjugated/unconjugated BAs were significantly elevated in the PBC group. They developed random forest models for etiologic classification using a combination of 46 and 6 BA-related variables, showing high diagnostic performance (AUROC > 0.8) for distinguishing etiologies; being specifically the AUROC for PBC of 0.975 with 46BAs and of 0.955 with 6BAs [95].

Regarding stratification of PBC patients, Hegade et al. found that patients with PBC-associated pruritus had elevated levels of GCA and GCDCA compared to asymptomatic PBC controls, suggesting that BA profiling may offer utility in monitoring disease phenotypes [91].

However, their utility extends beyond pruritus, involving also the assessment of disease progression risk. In this regard, Trottier et al. identified that total primary BAs, TCDCA, and GCA are significantly associated with PBC progression [32]. Ma et al. further supports this vision, but their study described that the levels GCDCA, glycochenodeoxycholic sulfate, and TDCA progressively increased with advancing liver disease, as defined by the Child-Pugh classification [94]. On the other hand, Jiang et al. proposed total BAs to platelet ratio (TPR) as a simple inexpensive and easily accessible non-invasive fibrosis diagnostic model in PBC. In their study, AUROC of the TPR was 0.771, higher than other non-invasive serological models [96]. Furthermore, 154 BA-relevant variables have been correlated with clinical indices in PBC patients, with platelet count and albumin showing the highest number of significant correlations [95].

Finally, Martínez-Gili et al. [97]. evaluated the ability of BAs to predict therapeutic response to UDCA. They showed that responders had higher fecal secondary and tertiary BAs than non-responders and lower urinary BA abundances, with the exception of 12-dehydrocholic acid, which was higher in responders and therefore proposed as a potential biomarker of therapy response. Responders also showed higher levels of 12 fecal BAs compared to non-responders. Furthermore, they identified that the fecal microbiota of poor responders to UDCA showed lower abundance of phyla with BA-deconjugation capacity (*Actinobacteriota/Actinomycetota*, *Desulfobacterota*, *Verrucomicrobiota*) compared to responders [97].

Noteworthy, many of the mentioned studies present small sample size and heterogeneous designs, thus limiting its standardization and further clinical implementation. Furthermore, most of them have described differences between groups and statistical correlations and only four studies have evaluated diagnostic performance as AUROC, specificity and sensitivity [93,94,95,96] (Table 1). Although promising, validation studies with larger sample size and real-world settings are sorely needed to test the utility of BA profiling as diagnostic, prognostic and treatment response biomarkers in PBC.

## 5. Current Limitations and Future Directions

PBC is a complex disease for which there are still more questions than answers. However, over the past decade, there has been a significant increase in research interest as well as in therapeutic alternatives, to the great benefit of patients living with this disease.

Although PBC is classically classified as a rare disease due to its low prevalence, it is important to recognize that its distribution varies across different geographic regions. While the incidence of PBC has remained stable over recent decades, its prevalence is clearly rising in Europe, the Americas, and the Asia-Pacific region [1]. Even within these regions, prevalence rates vary considerably, for instance, Eastern Europe shows markedly lower prevalence compared to Western or Southern Europe [98]. This variability in geographic distribution may also be associated with differences in disease presentation. For example, the prevalence of AMA positivity has been reported to be lower in PBC patients diagnosed in Latin America [99] compared to those diagnosed in Europe. These regional differences in the epidemiology and serological presentation of PBC likely result from a multifactorial interplay involving genetic predispositions, environmental exposures, and healthcare practices. On the genetic front, population-level variability in allelic frequency, particularly within the HLA locus, may influence both susceptibility to PBC and the production of hallmark autoantibodies, such as AMA [100]. Furthermore, environmental factors such as degree of industrialization, exposure to environmental toxins, and smoking, have been implicated in modulating disease risk. Accordingly, geographic disparities in such exposures may underline regional variability in disease incidence and prevalence [101]. Considering that many of the cited studies come from Western populations, given the regional differences described, it is important that such variability is considered when generalizing the utility of results and biomarkers. In any case, this increasing prevalence offers greater opportunities for broader and more inclusive research across diverse populations, which will advance our understanding of the disease.

On the other hand, despite all these advances and although there is substantial evidence supporting a major role for BA in the pathogenesis of PBC, few studies have consistently examined the BA profile in human samples from PBC patients. Moreover, it remains unclear whether alterations in BA composition preceded the development of autoimmunity or arisen as a consequence of immune-mediated tissue damage. Besides, current clinical guidelines advise keeping UDCA treatment up to one year in order to check for therapy response, then adding second line treatment if insufficient therapy response. However, the utility of BA profiling in predicting treatment response in PBC patients remains poorly understood. Both Elafibranor and Seladelpar have demonstrated a significant impact on BA metabolism, either by reducing serum BA [86,87] or levels of circulating C4 [78,82]. However, no further data are currently available regarding their utility in risk stratification for disease progression or in predicting lack of treatment response.

Therefore, it is essential to explore new strategies to monitor both disease progression and therapeutic response, with the goal of advancing precision medicine. In this context, BA profiling emerges as a highly promising approach.

## 6. Conclusions

PBC is an autoimmune cholestatic liver disease driven by immune-mediated bile duct destruction and disrupted BA homeostasis, which critically contributes to disease progression. UDCA improves bile flow and survival but fails in a significant subset of patients, prompting development of second-line BA-targeted therapies that modulate nuclear receptors and inflammatory pathways. Circulating BA profiles have emerged as promising non-invasive biomarkers to monitor disease activity and treatment response; however, their clinical utility remains limited by incomplete mechanistic understanding and validation. Addressing these gaps through integrative mechanistic studies and longitudinal clinical trials is essential to optimize BA-centered therapeutic approaches and advance personalized management of PBC.

## Figures and Tables

**Figure 1 ijms-26-07905-f001:**
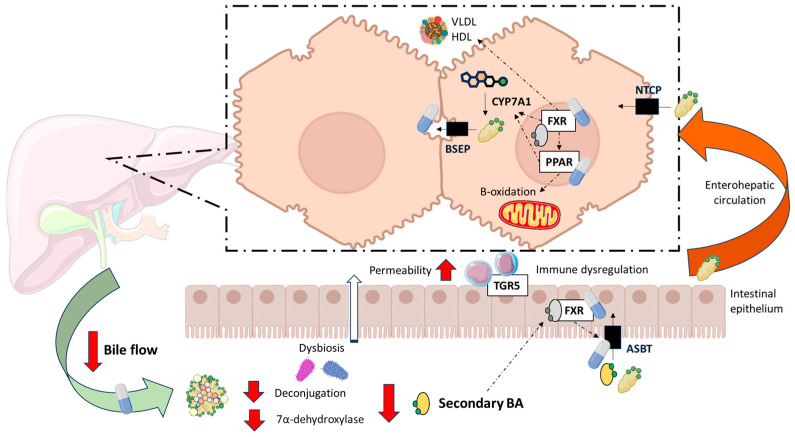
Alterations in bile acid and lipid metabolism in patients with Primary Biliary Cholangitis (PBC). Under physiological conditions, primary bile acids (BA) are synthesized in the liver from cholesterol. The cholesterol 7α-hydroxylase (CYP7A1) is the rate limiting enzyme. In the gut, primary BAs undergo deconjugation and 7α-dehydroxylation by anaerobic bacteria, resulting in the formation of secondary BA. In patients with PBC, bile flow from the liver to the intestinal lumen is reduced. There is also a reduction in deconjugation possibly associated with intestinal dysbiosis, increasing the primary to secondary BA ratio. As secondary BAs are the most potent agonists, its reduction could lead to dysregulation of Farnesoid X receptor (FXR) and G protein-coupled BA receptor (TGR5), increasing intestinal permeability and inflammatory responses. When FXR is dysregulated, homeostatic compensation of cholestasis cannot occur and therefore PBC progresses. Main pharmacological targets—FXR, Peroxisome Proliferator-Activated Receptors (PPARs), Apical Sodium-dependent Bile Acid Transporter (ASBT) and bile composition and flow—have been depicted. BSEP: Bile Salt Export Pump; HDL: High-Density Lipoprotein; NTCP: Na-Taurocholate Cotransporting Polypeptide. VLDL: Very-Low-Density Lipoprotein. Created with BioRender.com.

**Figure 3 ijms-26-07905-f003:**
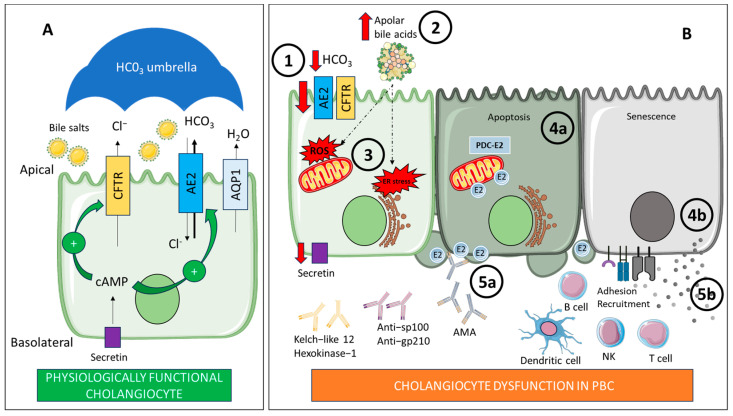
(**A**) Physiologically functional cholangiocyte: The cholangiocyte is a polarized cell with basolateral and apical membranes. On the apical membrane, bicarbonate (HCO_3_) secretion occurs via the Cl^−^/HCO_3_ exchanger AE2. This secretion is facilitated by the chloride gradient created by the CFTR channel. Secretin levels influence this secretion. Secretin increases intracellular cAMP levels, stimulating the transport of vesicles containing AE2, CFTR, and AQP1 to the apical membrane. This HCO_3_ secretion protects cholangiocytes by maintaining an alkaline pH barrier or “umbrella” over the outer surface of the apical membrane, keeping bile salts unprotonated. (**B**) Cholangiocyte dysfunction in PBC: There is a reduction in HCO_3_ secretion due to defective hepatic AE2 expression (1). This leads to the acidification of bile at the apical membrane, which promotes the persistence of an apolar state of glycine-conjugated bile acids (2). These bile acids can penetrate into cholangiocytes, causing oxidative and ER stress, apoptosis, and cytotoxicity (3). On the other hand, stressed cholangiocytes express immunogenic forms of PDC-E2 through apoptotic blebs (4a), which consequently activate and expand autoreactive B cells, resulting in the production of autoantibodies (5a). Furthermore, senescent cholangiocytes actively participate in the recruitment of immune cells, (4b) promoting the associated immune dysregulation (5b). AE2: Anion exchanger 2. AMA: Antimitochondrial antibodies. AQP1: Water channel aquaporin 1. B cell: B lymphocyte. cAMP: Cyclic adenosine monophosphate. CFTR: Cystic fibrosis transmembrane regulator. Cl^−^: Chloride. ER: Endoplasmic reticulum. HCO_3_: Bicarbonate. NK: Natural killer. PBC: Primary biliary cholangitis. PDC-E2: E2 component of pyruvate dehydrogenase complex. ROS: Reactive oxygen species. T cell: T lymphocyte. Created with BioRender.com and smart.servier.com.

**Table 1 ijms-26-07905-t001:** Performance of different serum bile acids (BA) and BA-related variables as biomarkers for diagnosis, differential diagnosis and disease progression in Primary Biliary Cholangitis.

Clinical Applicability	Biomarker	AUROC	Reference
Diagnosis	DCA	0.99	Liu et al. [93]
	GDCA	0.95	
	TUDCA	0.93	
	LCA	0.93	
	TLCA	0.91	
	TDCA	0.91	
	GUDCA	0.89	
	GCDCA	0.83	
Differential diagnosis	LCA	0.68	Ma et al. [94]
	LCA + TLCA	0.73	
	CDCA	0.74	
	TBA	0.52	
	46 BAs	0.97	Sang et al. [95]
	6 BAs	0.95	
Disease progression	TPR	0.77	Jiang et al. [96]

AUROC: Area Under the Receiver’s Operating characteristic Curve. DCA: Deoxycholic acid. GDCA: Glycodeoxycholic acid. TUDCA: Tauroursodeoxycholic acid. LCA: Lithocholic acid. TLCA: Taurolithocholic acid. TDCA: Taurodeoxycholic acid. GUDCA: Glycoursodeoxycholic acid. GCDCA: Glycochenodeoxycholic acid. CDCA: Chenodeoxycholic acid. TBA: Total bile acids. 46 BAs: Combination of 46 BA-related variables. 6 BAs: Combination of 6 BA-related variables. TPR: Total bile acid to platelet ratio.

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
