# Peer review of "Primary Biliary Cholangitis: Immunopathogenesis and the Role of Bile Acid Metabolism in Disease Progression"

_ijms, 2025, doi:10.3390/ijms26167905_

Round 1

Reviewer 1 Report

Comments and Suggestions for Authors

While I do realize the original report might appear too simplistic, my annotations are confirmed: it is a review, not a research paper and as such it is to be methodologically correct and factually accurate, and it is, with adequate iconography.
It does NOT add any significant new finding but does narratively condensate current knowledge on the task at hand and as such I deem it acceptable for publication.

-----------------------------------------------------

In this review, the Authors offer a narrative, concise review of the patophysiology of PBC, as well as its current treatment options and the role bile acids may play as markers, disease actors and treatment targets.

The review is well circumstantiated, accurate in its description and satisfactorily touches upon all the required aspects of the matter-at-hand. I thus have no particular points of concern with this work. 

Author Response

Comments and Suggestions for Authors

While I do realize the original report might appear too simplistic, my annotations are confirmed: it is a review, not a research paper and as such it is to be methodologically correct and factually accurate, and it is, with adequate iconography.
It does NOT add any significant new finding but does narratively condensate current knowledge on the task at hand and as such I deem it acceptable for publication.

In this review, the Authors offer a narrative, concise review of the patophysiology of PBC, as well as its current treatment options and the role bile acids may play as markers, disease actors and treatment targets.

The review is well circumstantiated, accurate in its description and satisfactorily touches upon all the required aspects of the matter-at-hand. I thus have no particular points of concern with this work. 

Response:

We sincerely thank the reviewer for their thoughtful and encouraging comments. We truly appreciate the recognition of our work as a concise and accurate review of the pathophysiology and treatment of PBC, as well as the acknowledgment of the methodological soundness and adequacy of the iconography. We are grateful for your positive assessment and for deeming the manuscript acceptable for publication. Your feedback is highly encouraging.

Reviewer 2 Report

Comments and Suggestions for Authors

I must acknowledge that this manuscript addresses a topic of significant clinical and scientific relevance - primary biliary cholangitis and the central role of bile acids in its pathogenesis. The authors demonstrate deep knowledge of the literature and present a comprehensive synthesis of the pathophysiological mechanisms involved. However, several areas emerge that require substantial improvements to elevate the overall quality of the work.

Structure and Content Organization

The overall structure of the manuscript follows an appropriate logic but presents some significant imbalances. The introductory section, while providing adequate clinical context, is excessively detailed in describing pharmacological treatments, shifting focus from the declared objective of the review. This emphasis on treatments would be more appropriate in a dedicated section or could be condensed to allow greater space for the central pathophysiological aspects. The pathophysiology section, which represents the heart of the manuscript, suffers from fragmentation that hinders understanding of integrated mechanisms. The authors treat autoimmunity, cholestasis, and biliary damage separately, but the interconnected nature of these processes does not emerge clearly. A more integrated approach that highlights how these mechanisms reciprocally influence each other in disease progression would be beneficial.

Scientific Quality and Methodological Rigor

The manuscript demonstrates a solid scientific foundation but presents some methodological gaps. No systematic description is provided of the bibliographic search strategy used, databases consulted, or study selection criteria. This lack is particularly problematic when the authors discuss conflicting evidence, such as cardiovascular risk in PBC, without providing clear criteria for evaluating the quality and relevance of cited studies. The discussion of bile acids as biomarkers presents promising data but lacks critical analysis of the methodological limitations of cited studies. Many of the mentioned studies present small samples and heterogeneous designs, aspects that should be discussed more explicitly to provide readers with a balanced evaluation of the current state of evidence.

Expository Clarity and Accessibility

The writing is generally clear, but some sections suffer from excessive terminological density that may hinder comprehension. The bile acid metabolism section, in particular, presents a cascade of abbreviations and compound names that could benefit from a more gradual and didactic presentation. Introduction of a summary table of the main bile acids and their characteristics could facilitate understanding. The provided figures are informative but could be optimized. Figure 1, while comprehensive, appears visually dense and could benefit from simplification or subdivision into more specific panels. Figure 2 is more effective but could be enriched with directional arrows that guide the reader through the sequence of pathological events.

Innovative Aspects and Scientific Contribution

The manuscript presents several innovative aspects, particularly the emphasis on the "bicarbonate umbrella" hypothesis as a unifying mechanism in PBC pathogenesis. However, this theory, while fascinating, is presented relatively superficially. The authors could deepen the therapeutic implications of this hypothesis and discuss in greater detail the experimental evidence supporting it. The discussion of bile acids as biomarkers represents a potentially significant contribution but needs greater systematization. It would be useful to organize this information in a table comparing sensitivity, specificity, and clinical applicability of the different proposed biomarkers, providing readers with a practical tool to evaluate their potential clinical use.

A significant gap concerns the limited discussion of ethnic and geographic differences in PBC prevalence and presentation. Considering that much of the cited studies come from Western populations, it would be appropriate to discuss the generalizability limitations of the results. The section on future therapeutic perspectives appears limited. Given the rapid development of new therapeutic approaches in PBC, a more in-depth discussion of emerging strategies based on understanding bile acid mechanisms would add value to the manuscript. The manuscript would benefit from restructuring that better integrates the different pathophysiological aspects into a unified model of disease progression. Creating a conceptual figure showing the temporal evolution of different pathological mechanisms could significantly improve reader comprehension. It is necessary to strengthen the critical discussion of current study limitations and clearly identify areas requiring further research. This would not only demonstrate scientific rigor but also provide clear directions for future studies.

This manuscript addresses a topic of great clinical importance with scientific competence and provides a useful synthesis of current knowledge. With the suggested modifications, particularly regarding integration of pathophysiological mechanisms and critical deepening of evidence, it could become a significant contribution to PBC literature. The focus on bile acids as a central element in pathogenesis and as potential biomarkers represents an innovative approach that, if developed more systematically, could positively influence research and clinical practice in this field. The manuscript demonstrates scholarly merit and addresses important clinical questions, but requires refinement in organization, critical analysis, and integration of concepts to reach its full potential as a comprehensive review of this complex disease.

Author Response

Reviewer 2

Comment 1:

I must acknowledge that this manuscript addresses a topic of significant clinical and scientific relevance - primary biliary cholangitis and the central role of bile acids in its pathogenesis. The authors demonstrate deep knowledge of the literature and present a comprehensive synthesis of the pathophysiological mechanisms involved. However, several areas emerge that require substantial improvements to elevate the overall quality of the work.

Response 1:

We are grateful for reviewer recognition of the manuscript’s scientific merit and clinical relevance. Their constructive recommendations have guided substantial improvements in the depth, organization, and clarity of the manuscript. Hereafter, we address point by point all reviewer’s comments and we have also attached a document highlining the changes performed in the manuscript to ease second revision.

Comment 2: Structure and Content Organization

The overall structure of the manuscript follows an appropriate logic but presents some significant imbalances. The introductory section, while providing adequate clinical context, is excessively detailed in describing pharmacological treatments, shifting focus from the declared objective of the review. This emphasis on treatments would be more appropriate in a dedicated section or could be condensed to allow greater space for the central pathophysiological aspects. The pathophysiology section, which represents the heart of the manuscript, suffers from fragmentation that hinders understanding of integrated mechanisms. The authors treat autoimmunity, cholestasis, and biliary damage separately, but the interconnected nature of these processes does not emerge clearly. A more integrated approach that highlights how these mechanisms reciprocally influence each other in disease progression would be beneficial.

Response 2:

We thank the reviewer for this insightful comment regarding the structure and content organization of the manuscript. In response, we have revised the text substantially to achieve a more balanced and integrated presentation of the material. Specifically, we have created a new Section 3, dedicated to the role of bile acids in the pharmacological treatment of primary biliary cholangitis (PBC). This change allows us to streamline the introductory section by relocating detailed descriptions of therapeutic agents to a more appropriate part of the manuscript.

Additionally, we have expanded the subsection addressing the “bicarbonate umbrella” mechanism, incorporating new evidence and updated references to better highlight its pathophysiological relevance (new lines 305-308, 310-314 and 318-320). These modifications aim to provide a more cohesive and comprehensive understanding of the interconnected processes of autoimmunity, cholestasis, and biliary epithelial damage in PBC.

-----------------------------------------------------------------------------------------------------------------

Comment 3: Scientific Quality and Methodological Rigor

The manuscript demonstrates a solid scientific foundation but presents some methodological gaps. No systematic description is provided of the bibliographic search strategy used, databases consulted, or study selection criteria. This lack is particularly problematic when the authors discuss conflicting evidence, such as cardiovascular risk in PBC, without providing clear criteria for evaluating the quality and relevance of cited studies. The discussion of bile acids as biomarkers presents promising data but lacks critical analysis of the methodological limitations of cited studies. Many of the mentioned studies present small samples and heterogeneous designs, aspects that should be discussed more explicitly to provide readers with a balanced evaluation of the current state of evidence.

Response 3:

We sincerely thank the reviewer for their insightful comments regarding the methodological aspects of our manuscript. As this is a narrative review, our aim was not to conduct a systematic analysis of the literature, but rather to provide an integrative overview of key pathophysiological processes in PBC, informed by recent advances in the field. Nonetheless, in response to the reviewer’s concerns, we have added a brief paragraph in the Introduction (New lines 49-51) clarifying the narrative nature of the review and specifying the main databases consulted (PubMed, Scopus, and Web of Science).

In addition, we have strengthened the discussion of methodological limitations in the sections addressing cardiovascular risk in PBC (New lines 241-242) and the use of bile acids as biomarkers (New lines 463-469). We now explicitly comment on key issues such as small sample sizes and heterogeneity of study designs, to better contextualize the interpretation of the available evidence and offer readers a more nuanced and balanced perspective.

We hope these additions contribute to enhancing the scientific transparency and critical appraisal of the manuscript.

----------------------------------------------------------------------------------------------------------------

Comment 4: Expository Clarity and Accessibility

The writing is generally clear, but some sections suffer from excessive terminological density that may hinder comprehension. The bile acid metabolism section, in particular, presents a cascade of abbreviations and compound names that could benefit from a more gradual and didactic presentation. Introduction of a summary table of the main bile acids and their characteristics could facilitate understanding. The provided figures are informative but could be optimized. Figure 1, while comprehensive, appears visually dense and could benefit from simplification or subdivision into more specific panels. Figure 2 is more effective but could be enriched with directional arrows that guide the reader through the sequence of pathological events.

Response 4:

We are grateful to the reviewer for their constructive feedback regarding expository clarity and visual accessibility. In response, we have implemented several changes to enhance the readability and didactic quality of the manuscript.

First, Figure 1 has been simplified to reduce visual complexity and improve conceptual clarity. We have also added a new figure (now Figure 2) that provides a concise summary of the main bile acids and their chemical characteristics, as suggested. This visual aid is intended to support the reader’s understanding of the bile acid metabolism section.

Furthermore, the new Figure 3 (previously Figure 2) has been enhanced with numbered elements, which are now explained in detail in the figure legend to guide the reader more effectively through the sequence of pathological events depicted.

----------------------------------------------------------------------------------------------------------------

Comment 5: Innovative Aspects and Scientific Contribution

The manuscript presents several innovative aspects, particularly the emphasis on the "bicarbonate umbrella" hypothesis as a unifying mechanism in PBC pathogenesis. However, this theory, while fascinating, is presented relatively superficially. The authors could deepen the therapeutic implications of this hypothesis and discuss in greater detail the experimental evidence supporting it. The discussion of bile acids as biomarkers represents a potentially significant contribution but needs greater systematization. It would be useful to organize this information in a table comparing sensitivity, specificity, and clinical applicability of the different proposed biomarkers, providing readers with a practical tool to evaluate their potential clinical use.

A significant gap concerns the limited discussion of ethnic and geographic differences in PBC prevalence and presentation. Considering that much of the cited studies come from Western populations, it would be appropriate to discuss the generalizability limitations of the results. The section on future therapeutic perspectives appears limited. Given the rapid development of new therapeutic approaches in PBC, a more in-depth discussion of emerging strategies based on understanding bile acid mechanisms would add value to the manuscript. The manuscript would benefit from restructuring that better integrates the different pathophysiological aspects into a unified model of disease progression. Creating a conceptual figure showing the temporal evolution of different pathological mechanisms could significantly improve reader comprehension. It is necessary to strengthen the critical discussion of current study limitations and clearly identify areas requiring further research. This would not only demonstrate scientific rigor but also provide clear directions for future studies.

This manuscript addresses a topic of great clinical importance with scientific competence and provides a useful synthesis of current knowledge. With the suggested modifications, particularly regarding integration of pathophysiological mechanisms and critical deepening of evidence, it could become a significant contribution to PBC literature. The focus on bile acids as a central element in pathogenesis and as potential biomarkers represents an innovative approach that, if developed more systematically, could positively influence research and clinical practice in this field. The manuscript demonstrates scholarly merit and addresses important clinical questions, but requires refinement in organization, critical analysis, and integration of concepts to reach its full potential as a comprehensive review of this complex disease.

 Response 5:

We thank the reviewer for highlighting the innovative potential of the manuscript.

As mentioned previously, we have expanded the subsection discussing the ‘bicarbonate umbrella’ mechanism, including a more detailed account of its pathophysiological relevance and potential therapeutic implications, supported by recent experimental evidence. Throughout the manuscript, we have strengthened the critical discussion of current study limitations.

Additionally, we have introduced a new Table 1, which provides a comparative summary of the available data on the diagnostic performance of bile acids and related variables as serum biomarkers for PBC. This table includes information on clinical applicability and AUROC where available. However, we also emphasize in the revised text that many studies do not report standardized performance metrics, which limits the ability to draw definitive conclusions. These limitations are now discussed explicitly in the corresponding section (New lines 463-469).

Finally, we fully agree with the observation regarding the limited discussion of ethnic and geographic variability in PBC. In response, we have expanded the conclusion section to address the differential prevalence and clinical presentation observed in different populations (new lines 482-490). We also discuss how the predominance of Western-based studies may affect the generalizability of the conclusions drawn, emphasizing the need for broader, more inclusive research across diverse populations. Furthermore, we have clearly identified areas requiring further research (New lines 491-495).

We want to thank again the reviewer for the in-depth revision and significant contribution to improve our manuscript. We are confident that these revisions significantly enhance the manuscript’s contribution to the field.

Round 2

Reviewer 2 Report

Comments and Suggestions for Authors

I appreciate the authors' comprehensive response to my previous comments and their substantial efforts to improve the manuscript. The revisions demonstrate a clear commitment to addressing the concerns raised and have significantly enhanced the overall quality of the work.

The creation of a dedicated Section 3 for pharmacological treatments effectively addresses the structural imbalance previously noted. The relocation of detailed therapeutic descriptions from the introduction allows for better focus on pathophysiological mechanisms. The expansion of the "bicarbonate umbrella" section with new evidence (lines 305-308, 310-314, 318-320) provides valuable mechanistic insights.

While I acknowledge this is a narrative review, the addition of the brief methodological paragraph (lines 49-51) and the enhanced discussion of study limitations (lines 241-242, 463-469) represent improvements. However, the critical analysis of conflicting evidence, particularly regarding cardiovascular risk, could still benefit from more systematic evaluation of study quality differences.

The simplified Figure 1 and the addition of the new Figure 2 summarizing bile acid characteristics are valuable didactic improvements. The enhanced Figure 3 with numbered elements and detailed legend significantly improves reader comprehension of the pathological sequence.

The addition of Table 1 providing comparative biomarker performance data is an excellent systematic contribution. The expanded discussion of geographic and ethnic variability (lines 482-490) addresses an important generalizability concern. The identification of future research areas (lines 491-495) provides clear direction for the field.

Remaining Concerns and Suggestions

  1. While the new figures represent improvements, ensure that Figure 2 maintains consistent formatting with the rest of the manuscript and that all abbreviations are clearly defined in the legend.

  2. Consider adding confidence intervals for AUROC values where available, and briefly discuss why many studies lack standardized performance metrics.

  3.  The added content on regional differences is valuable but could benefit from a brief discussion of potential underlying causes (genetic, environmental, diagnostic practices).

This revised manuscript demonstrates significant improvement in organization, critical analysis, and scientific rigor. The authors have successfully transformed the work into a comprehensive review that will serve as a valuable resource for clinicians and researchers in the field. The systematic approach to bile acid biomarkers and the integration of therapeutic mechanisms with pathophysiology are particularly commendable.

The manuscript is now suitable for publication pending minor revisions to address the remaining suggestions outlined above.

Author Response

We sincerely appreciate the time and effort you have dedicated to reviewing our manuscript.

We have meticulously addressed each of the points raised by the reviewer. Below is a detailed response to the comments, which we hope will meet your expectations:

Reviewer 2 Comment 1: “While the new figures represent improvements, ensure that Figure 2 maintains consistent formatting with the rest of the manuscript and that all abbreviations are clearly defined in the legend”

Response: Following your suggestions, part of the legend of Figure 2 has been modified. However, abbreviations regarding the different types of bile acids have been included in the main figure to ease the understanding of the text, so they are clearly visible and they can be checked while reading the main text.

Reviewer 2 Comment 2: “Consider adding confidence intervals for AUROC values where available, and briefly discuss why many studies lack standardized performance metrics”

Response: Thank you very much for your suggestion. In the section "Bile acids as biomarkers of PBC progression and therapy response", confidence intervals have been searched for but were not available in the original articles. However, the AUROCs have been specified more clearly in the main text.

Reviewer 2 Comment 3: “The added content on regional differences is valuable but could benefit from a brief discussion of potential underlying causes (genetic, environmental, diagnostic practices).”

Response: As suggested, information about genetic differences, environmental influences, and geographical variations has been added to the conclusions section.

We are truly grateful for your constructive feedback and the opportunity to improve our manuscript based on your expert recommendations. Please find the revised version of the manuscript attached for your consideration.
